

# Genome-wide identification and characterization of *SPX domain-containing* genes family in eggplant

Li Zhuomeng[1], Tuo Ji[1,2,3], Qi Chen[1], Chenxiao Xu[1], Yuqing Liu[1], Xiaodong Yang[4], Jing Li[1,2,3] and Fengjuan Yang[1,2,3]

[1] College of Horticulture Science and Engineering, Shandong Agricultural University, Tai an, China
[2] Key Laboratory of Biology and Genetic Improvement of Horticultural Crops (Huanghuai Region), Ministry of Agriculture and Rural Affairs, Shandong Agricultural University, Tai an, China
[3] Shandong Collaborative Innovation Center for Fruit and Vegetable Production With High Quality and Efficiency, Shandong Agricultural University, Tai an, China
[4] Weifang Academy of Agricultural Science, Weifang, China

Corresponding authors
Jing Li, lij19900525@163.com
Fengjuan Yang, beautyyfj@163.com

## ABSTRACT

Phosphorus is one of the lowest elements absorbed and utilized by plants in the soil. *SPX domain-containing* genes family play an important role in plant response to phosphate deficiency signaling pathway, and related to seed development, disease resistance, absorption and transport of other nutrients. However, there are no reports on the mechanism of *SPX domain-containing* genes in response to phosphorus deficiency in eggplant. In this study, the whole genome identification and functional analysis of *SPX domain-containing* genes family in eggplant were carried out. Sixteen eggplant *SPX domain-containing* genes were identified and divided into four categories. Subcellular localization showed that these proteins were located in different cell compartments, including nucleus and membrane system. The expression patterns of these genes in different tissues as well as under phosphate deficiency with auxin were explored. The results showed that *SmSPX1*, *SmSPX5* and *SmSPX12* were highest expressed in roots. *SmSPX1*, *SmSPX4*, *SmSPX5* and *SmSPX14* were significantly induced by phosphate deficiency and may be the key candidate genes in response to phosphate starvation in eggplant. Among them, *SmSPX1* and *SmSPX5* can be induced by auxin under phosphate deficiency. In conclusion, our study preliminary identified the SPX domain genes in eggplant, and the relationship between *SPX domain-containing* genes and auxin was first analyzed in response to phosphate deficiency, which will provide theoretical basis for improving the absorption of phosphorus in eggplants through molecular breeding technology.

# INTRODUCTION

Phosphorus (P) is an essential nutrient for plant development and reproduction (*Hou et al., 2020*; *Mng'ong'o et al., 2021*; *Ojeda-Rivera et al., 2022*; *Wang et al., 2022*), which is involved

in photosynthesis, respiration, energy storage and transfer, cell division and cell enlargement in plants. Low utilization of phosphate (P) in agricultural soils has seriously damaged crop production, resilience, seed germination and soil biological activity (*Prathap et al., 2022*; *Suriyagoda et al., 2023*; *Wang et al., 2022*). Large-scale application of phosphate fertilizer is the most common and expensive way to overcome the problem, which brings the waste of phosphate rock. Furthermore, plants could only absorb 10–15% of the total fertilizer applied, more than 80% of the fertilizer cannot be used because of soil absorption (*Prathap et al., 2022*). How to reduce fertilizer dosage, environmental pollution and improve plant productivity is a great challenge to be overcome in the future. Therefore, it is an important research direction for the development of agricultural industry to improve phosphorus utilization rate (*Hou et al., 2020*). Plants respond to phosphate deficiency stress mainly by changing root system architecture (RSA), controlling rhizosphere P activation, and participating in phosphorus recovery and conservation. Previous studies have reported that the response of root remodeling to phosphate deficiency was mainly regulated by genes and plant hormones (*Katznelson, 1977*; *Satheesh et al., 2022*).

The *SPX domain-containing* protein family was named by three members: *SYG1, PHO81,* and *XPR1*. They play a main function in maintaining P homeostasis at the cellular level through P transport and adaptation to P deficiency (*Jung et al., 2018*; *Liu et al., 2018*). *SPX domain-containing* family genes have been identified in many plant species, such as 20 in *Arabidopsis* (*Duan et al., 2008*), 33 in *Zea mays* (*Xiao et al., 2021*), 23 in *Phyllostachys edulis*, 46 in the *Triticum*, 69 in the *Brassica napus* (*Luo et al., 2023*; *Yang et al., 2022*). They could be divided into four different subfamilies in plants: SPX proteins subfamily only containing SPX domain, SPX–EXS proteins subfamily containing SPX and an EXS (*ERD1*, *XPR1* and *SYG1*) domain, SPX–MFS proteins subfamily containing SPX and the major facility superfamily (MFS) domain, and SPX–RING proteins subfamily containing SPX and the RING-type zinc finger domain (*Kant, Peng & Rothstein, 2011*; *Liu et al., 2018*; *Wang et al., 2012*). The SPX subfamily functions primarily as phosphate-sensing signaling proteins. AtSPX1, AtSPX2, and AtSPX4 are functionally redundant in *Arabidopsis* and act as a negative regulator of the p-signaling center-regulated gene *phosphate starvation response 1* (*AtPHR1*) in different P concentration (*Zhou et al., 2015*). AtSPX1 interacts with AtPHR1 to inhibit the phosphoric acid starvation induction (PSI) gene through P1BS, resulting in reducing expression of the phosphoric acid starvation response (PSR) gene. In the condition of P deficiency, the interaction between AtSPX1/AtPHR1 is weakened, which promotes the binding of AtPHR1 to P1BS and regulates the expression of PSR gene. Inhibition of *AtSPX3* can aggravate the symptoms of phosphate deficiency, change P distribution and enhance the expression of *PSRs* (*Puga et al., 2014*). In maize, ZmSPX3, ZmSPX4.2, ZmSPX5, and ZmSPX6 can interact with ZmPHR1 respectively, to participate in the response to phosphate deficiency stress in a ZmPHR1-mediated manner (*Xiao et al., 2021*). SPX-MFS mainly transfers phosphate into vacuoles, one of the SPX-MSF members, *VPT1* (*vacuolar phosphate transporter 1*), also known as *PHT5; 1* (*phosphate transporter 5; 1*), plays a major role on P sequestration in *Arabidopsis* vacuoles. The *vpt1* mutants showed sensitivity to P stress, which supported that *VPT1* was essential for *Arabidopsis thaliana* to adapt to phosphate deficiency stress (*Liu et al., 2016a*; *Młodzińska & Zboińska, 2016*).

The phosphate transporter *PHT1* is responsible for taking up phosphate from the soil and further distributing it to aboveground plant organs, the expression of *PHT1* was induced or strongly up-regulated during P deprivation (*Młodzińska & Zboińska, 2016*; *Zhang et al., 2015*). *OsSPX-MFS1* is a key player in the maintenance of P homeostasis in leaves and may act as a P transporter (*Lin et al., 2010*). Both *OsSPX-MFS1* and *OsSPX-MFS2* are negatively regulated by *osa-miR827* abundance in response to phosphate starvation (*Zhang et al., 2015*). In wheat, the *TaSPX-MFS* subfamily genes were targeted by nine different miRNAs, including *Tae-miR1120A, Tae-miR1120b-3p, Tae-miR1120b-5p, Tae-miR1122c-3p, Tae-miR1122a, Tae-miR3b-1130p, Tae-miR1130a, and Tae-miR3b-1137p*, and P starvation significantly induced the *TaSPX* gene expression (*Kumar et al., 2019*; *Kumar et al., 2018*). In *Brassica napus*, one *SPX-MFS* subfamily 11 genes were significantly induced by P starvation and recovered rapidly after P re-feeding (*Du et al., 2017*; *Yang et al., 2022*). SPX-EXS family plays an important role in phosphate acquisition, translocation, and distribution, mainly transfers phosphate into vascular cylinders in root, leaf, stem, or flower tissue (*Liu et al., 2019*; *Wang et al., 2004*). *SlPHO1;1* is a gene in the SPX-EXS subfamily of tomatoes, which plays a major role in phosphate transport between roots and stems at seedling stage (*Li, You & Zhao, 2021*; *Liu et al., 2019*; *Wang, Secco & Poirier, 2008*). The SPX-RING subfamily recognizes PHR1 ubiquitination PHR1 complexes through the SPX domain. *SYG1(NLA/BAH1)* is the only one SPX-RING gene identified in *Arabidopsis*, which was first reported to have an E3 ubiquitin ligase and respond to nitrogen-restricted transcription (*Kosarev, Mayer & Hardtke, 2002*; *Peng et al., 2007*). It was later found that *SYG1* was not only a target of *microRNA 827*, but also involved in the synthesis of salicylic acid and plant disease resistance (*Kant, Peng & Rothstein, 2011*; *Val-Torregrosa et al., 2022*; *Yaeno & Iba, 2008*). Recent studies have found that *SYG1* act as a negative regulator of *PHR1* by ubiquitylation, and inositol polyphosphate promotes the interactions between SYG1 and PHR1 resulting in the PHR1 destruction; overexpression of *SYG1* seedlings showed accumulation of long root hairs and anthocyanins in the shoot under P deficiency (*Park et al., 2023*).

Auxin regulating root development relies heavily on the transcription of auxin response genes in (Aux/IAA)—auxin response factor (ARF) auxin signaling modules (*Goh et al., 2012*; *Yang et al., 2021*). P deprivation increases the expression of *TIR1*, which encoding an auxin receptor that mediates auxin-regulated transcription, in *Arabidopsis* seedlings and enhances auxin sensitivity in P-deficient plants, thereby accelerating the degradation of AUX/IAA proteins and unshackling ARF transcription factors to activate/repress genes involved in lateral root formation and emergence (*Pérez-Torres et al., 2008*). Mutations that disrupt auxin synthesis (*taa1*) and transport (*aux1*) can inhibit the generation of root hairs in *Arabidopsis*. The expression level of *ARF19*, *RSL2*, and *RSL4* induced by auxin under phosphate deficiency conditions enhanced in root hair, and the lack of these genes can disrupt root hair generation (*Satheesh et al., 2022*). Additionally, the transcription factors ARF7 and ARF19 in *Arabidopsis* roots positively regulate the phosphorus deficiency response gene *PHR1* (*Huang et al., 2018*), while the *arf7* and *arf19* double mutant plants show a significant reduction in the expression of certain PSI genes. It has been reported that *OsPHR2* regulates the downstream gene *OsSPX1* in rice, suggesting potential regulatory

effects of auxin on some *SPX* genes (*Huang et al., 2018*; *Wang et al., 2009*). *SPX domain-containing* genes and auxin are both members that play a role under phosphate deficiency stress. Previous studies have reported that auxin could up-regulated *AtPHO1;H1* and *AtPHO1;H10* under phosphate deficiency (*Ribot, Wang & Poirier, 2008*). However, the relationship between other *SPX domain-containing* subfamilies genes and auxin is still unclear.

As a world's cash crop with a vast area of cultivation, Eggplant, contains rich anthocyanins and is a vegetable with high nutritional value (*Añibarro Ortega et al., 2022*). The absorption of phosphorus by eggplants mainly occurs during the flowering and fruiting stages (*Li et al., 2019*), indicating that phosphorus is one of the important factors determining the formation of eggplant yield. Howerver, phosphorus is one of the lowest elements absorbed and utilized by plants in the soil (*Katznelson, 1977*). There is currently no report on the mechanism of eggplant response to phosphate deficiency stress. In this study, sixteen potential *SPX domain-containing* genes were identified and their phylogeny, gene structure and conserved motifs were analyzed. We observed the subcellular localization of eight *SPX domain-containing genes* and determined their gene expression patterns in different tissues as well as under phosphate deficiency with auxin. These results will be helpful for improving the absorption of phosphorus in eggplants through molecular breeding.

## METHODS

### Identification of SPX domain-containing proteins in *Solanum melongena*

To identify proteins with SPX domains in the eggplant genome, ''SPX'' was used to search the TAIR (https://www.Arabidopsis.org/) database. Then, the *SPX domain-containing* biomolecular structure in *Arabidopsis thaliana* were screened and the SGN (https://solgenomics.net/) (*Barchi et al., 2021*) database was searched by BLASTP. Submit the detected biomolecular structure to the Pfam database (http://pfam.xfam.org/) (*El-Gebali et al., 2019*) and the SMART web site (http://SMART.embl-heidelberg.de/) (*Letunic & Bork, 2018*) to identify SPX domain. *Solanum lycopersicum* contains 19 SPX domain proteins (Table S2) (*Li, You & Zhao, 2021*).

### SPX domain-containing protein properties, subcellular localization prediction, gene structure analysis

Each biomolecular structure containing the SPX domain was submitted to ExPaSy (http://expasy.org/) (*Wilkins et al., 1999*) for calculation of molecular weight (MW) and isoelectric point (pI). The gene number, coding region length (CDS) and amino acid length (ORF) of each SPX domain-containing protein were obtained from SGN (https://solgenomics.net/) (*Barchi et al., 2021*) database. For predicting subcellular localization of proteins, we used (TMHMM-2.0) to predict the transmembrane structure of 16 SmSPX proteins. The co-linear images between the untranslated regions of all *SPX domain-containing* genes and the coding sequences between the *SPX domain-containing* genes of multiple species were drawn using TBtools (software version 1.132) (*Chen et al., 2020*).

## Chromosome information, phylogenetic tree construction and domain analysis of *SPX domain-containing* genes

The collinearity between the chromosome position information of SPX domain genes and the genes was generated by TBtools (software version 1.132) (*Chen et al., 2020*). The phylogenetic tree was built using MEGA 11.0 (*Tamura, Stecher & Kumar, 2021*) software and the neighbour-joining (NJ) method with 1,000 bootstrap replicates. The same method was applied to construct phylogenetic trees containing *SPX domain-containing* genes in different species. The high-quality phylogenetic tree graph of different species was optimized by online website ITOL (https://itol.embl.de/). SPX domain protein family domain information was identified using the online site Pfam (http://pfam-legacy.xfam.org/) (*El-Gebali et al., 2019*), and domain maps were generated using TBtools (software version 1.132) (*Chen et al., 2020*). Conserved motifs were identified using MEME software version 5.5.3 (https://meme-suite.org/tools/meme) with the following parameters: any number of repetitions, a maximum of 10 misfits, and an optimum motif width of 6–200 amino acid residues (*Bailey et al., 2009*).

## Subcellular localization

The complete coding sequences of eight *SPX domain-containing* protein genes were obtained from the SGN (https://solgenomics.net/) database (*Barchi et al., 2021*). CaMV35S promoter vector pRI 201 (Takara, Beijing, China) with GFP tag was linearized with NDEI endonuclease. Sequences of eight *SPX domain-containing* protein genes were combined into the vector using in-fusion Snap Assembly cloning kits (Takara, Dalian, China) according to the manual. The fusion construct was transferred into *Agrobacterium tumefaciens* strain LBA4404, and subcellular localization assays were performed as previously reported (*Sparkes et al., 2006*). The protein localization was determined under $20\times$ confocal microscopy (LSM880; Zeiss, Oberkochen, Germany). The primer sequences were listed in Table S1.

## Growth conditions and treatments

All seeds (V8 and NO.41 Multigenerational inbred lines) were soaked in gibberellin 60 mg/L for 4 h and then bathed in 55 °C water for 30 min, the seeds were washed twice and placed in a glass culture dish containing wet filter paper to germinate at 28 °C in constant darkness for 2 days. Seeds were grown at $25 \pm 2$ °C under a light regimen of 16 h light and 8 h dark at a temperature of $16 \pm 2$ °C for 40 days. The hydroponic solution was fed with the classic Hoagland 1/2 concentration nutrient solution, and the seedlings were starved of phosphate and normal phosphate after 2 days. Samples were taken at the end of the two-day period as 0-day control samples, and the samples were taken at the 1st, 5th and 10th day after phosphate starvation treatment. The difference between normal P and low P treatment was that the concentration of P in normal Group was 500 µM. The concentration of P was 10 µM in low-phosphate treatment group. The deficient K element was supplemented with $K_2SO_4$. The other components are identical. The concentration of IAA in low phosphate treatment was 0.2 mg/L, and IAA was dissolved with DMSO and dissolved directly in nutrient solution. Samples were taken at 1 day, 5 days and 10 days after phosphate starvation treatment with IAA.

## RNA extraction and qRT–PCR analysis

The process of collecting the sample data is similar to we conducted it previously (*Chen et al., 2022*). The tissue samples were flash-frozen in liquid nitrogen and subsequently stored at −80 °C for RNA extraction purposes. Total RNA was extracted from flowers, leaves, fruits, seeds, roots, and stems using RNA Isolation Kit (CWBIO) and then stored at −80 °C. The integrity of the extracted RNA was evaluated using the Agilent 2100 Bioanalyzer (Agilent Technologies, Santa Clara, CA, USA). First strand cDNA synthesis was performed on 1 μg of total RNA using the Prime Script RT Reagent Kit (Monad, Beijing, China). PCR amplification was carried out utilizing the ABI 7500 Fast Real-Time PCR system (Applied Biosystems, Foster City, CA, California, USA) and the ChemoHS Specificity Plus qPCR Mix Kit (Monad China). The amplification parameters were 95 °C for 10 min, followed by 40 cycles at 95 °C for 10 s, 60 °C for 30 s, and 72 °C for 10 s. The mRNA expression levels were normalized to the level of *SmActin* expression using the $2^{-\Delta\Delta Ct}$ method (*Livak & Schmittgen, 2001*). Each experiment included three biological replicates. The primer sequences are listed in Table S1.

## Statistical analysis

The error bar in all charts represents the standard deviation, we used SPSS 23.0 (SPSS, Inc., Chicago, IL, USA) and used one–way ANOVA and Duncan's New Multiple Range test ($P < 0.05$) to assess statistical significance.

# RESULTS

## Identification of SPX-domain-containing proteins in *S. melongena*

Based on the SGN (https://solgenomics.net/) database, the Pfam database (http://pfam.xfam.org/), the TAIR (https://www.Arabidopsis.org/) database, and the SMART software (http://SMART.embl-heidelberg.de/), a total of 16 SPX domain-containing proteins were identified in the eggplant genome (Table 1). Sixteen *SPX domain-containing* genes are divided into four subfamilies, including SPX, SPX-EXS, SPX-MFS and SPX-RING. Based on the subfamily and the chromosomes where the genes are located, we named the identified 16 *SPX domain-containing* genes from *SmSPX1* to *SmSPX16*. *SmSPX1–SmSPX5* belong to the SPX subfamily, *SmSPX6–SmSPX9* belong to the SPX-MFS subfamily, *SmSPX10–SmSPX14* belong to the SPX-EXS subfamily, and *SmSPX15–SmSPX16* belong to the SPX-RING subfamily. The coding sequences length of the 16 *SPX domain-containing* genes ranged from 780 bp to 4752 bp, and the amino acid sequences length ranged from 259 aa to 1583 aa. The predicted molecular weight ranged from 29.28KDa to 183.38KDa, and the isoelectric point ranged from 4.69 to 9.41.

## Chromosome location and collinearity analysis of *SPX domain-containing* genes in *S. melongena*

TBtools software was used to analyze all *SPX domain-containing* genes for chromosome localization and gene collinearity and mapping. There are 12 chromosomes in eggplant, and each chromosome contains *SPX domain-containing* genes. There are four SPX structural genes on chromosome 2. Chromosomes 1, 9, and 10 all contain two SPX structural genes,

**Table 1  SPX domain-containing gene family members in eggplant.** Members of the gene family containing the SPX domain in eggplant, gene length, homologous genes with other species, and chemical properties of foxes.

| Rename | Gene ID | Description | ORF (aa) | CDS (bp) | MW (KDa) | pI |
|--------|---------|-------------|----------|----------|----------|-----|
| SmSPX1 | SMEL4.1_01g017740.1.01 | SPX | 259 | 780 | 29.28 | 7.64 |
| SmSPX2 | SMEL4.1_02g009290.1.01 | SPX | 346 | 1041 | 38.5 | 5.35 |
| SmSPX3 | SMEL4.1_02g028380.1.01 | SPX | 302 | 909 | 34.25 | 4.69 |
| SmSPX4 | SMEL4.1_03g024350.1.01 | SPX | 290 | 873 | 33.26 | 5.99 |
| SmSPX5 | SMEL4.1_06g020500.1.01 | SPX | 266 | 801 | 31.24 | 6.2 |
| SmSPX6 | SMEL4.1_01g016630.1.01 | SPX-MFS | 575 | 1728 | 65.15 | 6.3 |
| SmSPX7 | SMEL4.1_05g022980.1.01 | SPX-MFS | 696 | 2091 | 77.98 | 6.03 |
| SmSPX8 | SMEL4.1_08g001320.1.01 | SPX-MFS | 694 | 2085 | 77.77 | 6.15 |
| SmSPX9 | SMEL4.1_08g022180.1.01 | SPX-MFS | 697 | 2094 | 78.36 | 5.94 |
| SmSPX10 | SMEL4.1_02g028360.1.01 | SPX-EXS | 1583 | 4752 | 183.38 | 9.27 |
| SmSPX11 | SMEL4.1_02g028370.1.01 | SPX-EXS | 629 | 1890 | 72.22 | 9.41 |
| SmSPX12 | SMEL4.1_09g022670.1.01 | SPX-EXS | 790 | 2373 | 91.04 | 9.23 |
| SmSPX13 | SMEL4.1_10g015110.1.01 | SPX-EXS | 791 | 2376 | 92.48 | 9.35 |
| SmSPX14 | SMEL4.1_10g017040.1.01 | SPX-EXS | 702 | 2109 | 82.05 | 9.18 |
| SmSPX15 | SMEL4.1_09g018170.1.01 | SPX-RING | 332 | 999 | 37.9 | 8.9 |
| SmSPX16 | SMEL4.1_12g008960.1.01 | SPX-RING | 325 | 978 | 37.37 | 8.04 |

and each of the remaining chromosomes contains only one SPX structural gene. In addition, an intergenic collinearity analysis of these 16 genes revealed that 37.5% (six of 16) of the *SPX domain-containing* genes were derived from gene duplication (Fig. 1A). Intergenic collinearity was mainly located in the chromosome 2 and chromosome 8, accounting for 66.7% of tandem repeats. Of these, two gene tandem repeats belong to the SPX subfamily and four gene tandem repeats belong to the SPX-MFS subfamily (Fig. 1B). By analyzing the genetic collinearity between *Arabidopsis*, *Solanum lycopersicum* and *S. melongena* we found that tomato and eggplant have high homology (Fig. 1C). Analysis of the isograms among eggplant, *Arabidopsis* and tomato showed that 11 genes from *Arabidopsis* had gene tandem events with eight genes from eggplants, and 12 genes from eggplant had gene tandem events with 12 genes from tomato, there were seven gene cascades in the three species.

## Phylogenetic tree analysis and gene structure of *SPX-domain-containing* genes in *S. melongena*

In order to analyze the evolutionary relationship of *SPX domain-containing* genes, a phylogenetic tree was constructed among *Arabidopsis*, *Zea mays L*, *Oryza sativaL*, *Solanum lycopersicum* and *S. melongena* (Fig. 2A). The phylogenetic tree divided into four subfamilies (SPX, SPX-MFS, SPX-EXS, SPX-RING). The first subfamily contains only SPX domains, including four *AtSPXs*, seven *ZmSPXs*, six *OsSPXs*, seven *SlSPXs*, and five *SmSPXs*. The second subfamily contains SPX and MFS domains, including three *AtSPX-MFSs*, 15*ZmSPX-MFSs*, three *OsSPX-MFSs*, four *SlSPX-MFSs* and four *SmSPX-MFSs*. The third subfamily of SPX and EXS domains consist of 11 *AtSPX-EXSs*, nine *ZmSPX-EXSs*, four *OsSPX-EXSs*, eight *SlSPX-EXSs* and five *SmSPX-EXSs*. The fourth subfamily contains SPX and RING
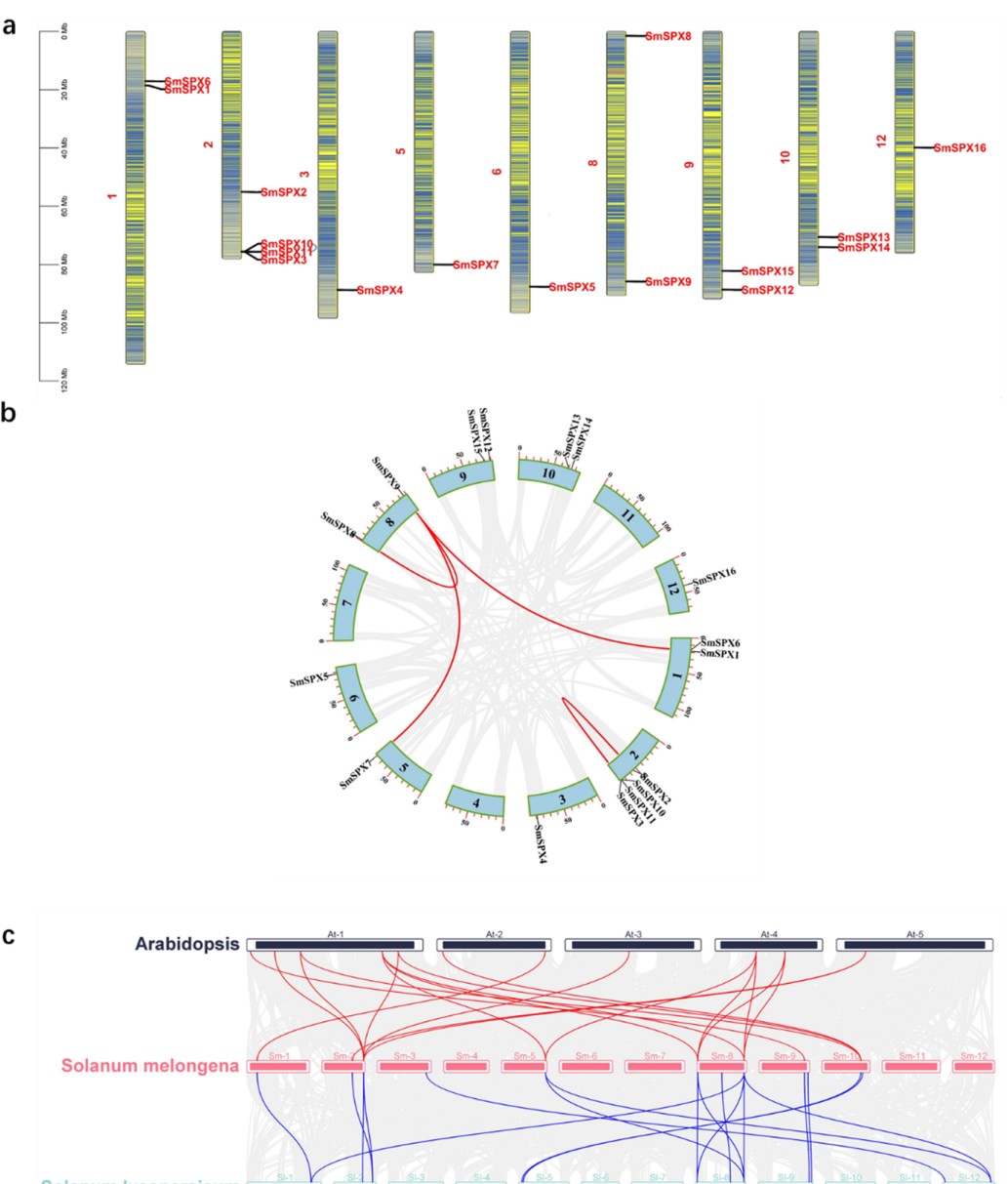

**Figure 1** (A) Physical map of 16 SPX-domain-containing genes in 12 chromosomes. The blue arcs in the graph correspond to genes that undergo tandem duplication. (B) Segmental duplication of the 16 SPX genes in 12 chromosomes. Genes linked with a line show a pair of segmentally duplicated genes. (C) Three species SPX gene collinearity analysis, red line indicates *Arabidopsis thaliana* and tomato gene duplication, blue line indicates tomato and eggplant gene duplication.

domains, including two *AtSPX-RINGs*, two *ZmSPX-RINGs*, two *OsSPX-RINGs*, two *SlSPX-RINGs* and two *SmSPX-RINGs*. In order to determine the motif composition of the *SPX-containing domain* genes, the conserved motif was analyzed by MEME online software, and a total of 10 conserved motifs were identified (Figs. 2B–2C), named as motif 1~10. SPX domain is located at the N-terminus, and SmSPX11 contains two sets of duplicate

SPX-EXS domains. Analyzing the gene structure of 16 *SPX-containing domain* genes (Fig. 2C) found that the SPX-EXS subfamilies have more complex exons.

## Subcellular localization of *SPX domain-containing* proteins

To better understand the function of the 16 *SPX-containing domain* genes, we conducted subcellular localization prediction. TMHMM-2.0 (https://services.healthtech.dtu.dk/services/TMHMM-2.0/) software was used to predict the transmembrane structure existed in the 16 *SPX-containing domain* genes. The results showed that *SmSPX6-14* had a transmembrane domain, and *SmSPX15-16* had no transmembrane domain, but it was expressed in the membrane system. Furtherly, the palmitoylation site of the SPX gene was predicted using (http://lipid.biocuckoo.org/index.php), and the signal appearing on the membrane may be due to the presence of palmitoylation sites in *SmSPX15* and *SmSPX16* (Table S3).

According to the analysis result of evolutionary relationship, the identified *SPXs* could be divided into four four subfamilies. Considering *SPXs* from the same subfamily might have similar function and localization, we selected randomly selected two members from each of the four subfamilies for subcellular localization. The selected genes were overexpressed using the *CaMV35S* promoter to form the target gene-GFP fusion protein. *35S::GFP* was used as positive control. We used cell membrane and nucleus reference (FM4-64, DAPI) to ensure the accuracy of localization (Fig. S1A and Fig. 3). *SmSPX1* and *SmSPX5* were detected in nucleus, *SmSPX6*, *SmSPX8*, *SmSPX10* and *SmSPX13* were detected in membrane system, *SmSPX15* and *SmSPX16* were detected in nucleus and membrane system.

## Tissue-specific expression of *SPX domain-containing* genes in *S. melongena*

Under phosphate deficiency condition, the root of the plant first senses and transmits the stress signal to the whole plant through complex signal transduction modes. To explore the tissue expression patterns of the 16 *SPX domain-containing* genes in eggplant, the mRNA was extracted from leaves, stems, roots, flowers, fruits, and seeds for qRT-PCR analysis (Fig. 4). *SmSPX1*, *SmSPX4*, *SmSPX5*, *SmSPX6*, *SmSPX12* and *SmSPX15* are expressed in roots. *SmSPX2*, *SmSPX3*, *SmSPX7*, *SmSPX9*, *SmSPX11* and *SmSPX13* were more expressed in stems, while *SmSPX8* and *SmSPX10* were more expressed in leaves. *SmSPX16* is specifically expressed in flowers, and all genes are little expressed in fruits, while *SmSPX1*, *SmSPX8* and *SmSPX11* are expressed in seeds.

## Expression patterns of 16 *SPX domain-containing* genes under phosphate deficiency stress

To identify the key genes under phosphate deficiency stress, we made root transcriptomes (PRJNA1030332, SAMN39278703–SAMN39278741) of two varieties with different phosphate deficiency tolerance treating with low phosphate for 1 day, 5 days, and 10 days. The Log2FKPM (fragment per thousand base transcripts per million mapped fragments) method was used to estimate the expression level of the *SPX domain-containing* genes, and the zero to one method was used to normalize the data. The 16 *SPX domain-containing*

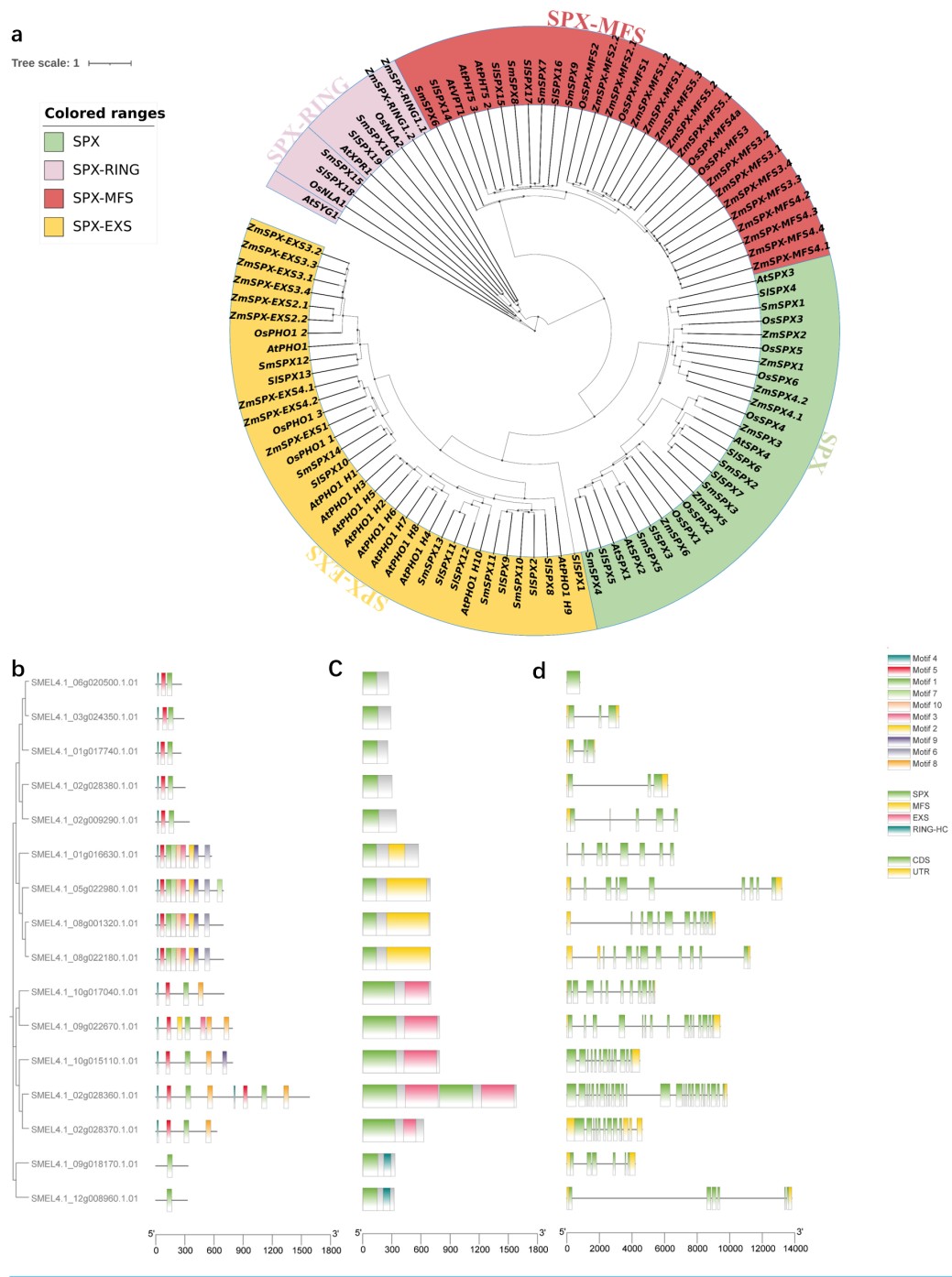

**Figure 2** Phylogenetic tree analysis and gene structure of *SPX-domain-containing* genes in *Solanum melongena.* (A) Phylogenetic analysis of SPX-domain-containing protein from *Arabidopsis*, maize, rice, tomato and eggplant. All genes are divided into four parts and labeled with different colors and text. Green background is SPX type, red is SPX-MFS type, yellow is SPX-EXS type, and pink is SPX-RING type. Phylogenetic relationships, conserved motifs, domains and gene structures of 16 SPX domain-containing genes. (continued on next page...)

**Figure 2 (...continued)**
(B) The phylogenetic relationships of the 16 *SPX domain-containing genes* constructed with bootstrap values of 1000 replicates. Conserved motifs and domains of 16 *SPX domain-containing* genes were analyzed using MEME. (C) Rounded rectangles with different colors indicate different domains. (D) The gene structures of the 16 *SPX domain-containing* genes analyzed using TBtools. CDS and UTRs are colored with green and yellow boxes, respectively, while black lines represent introns.

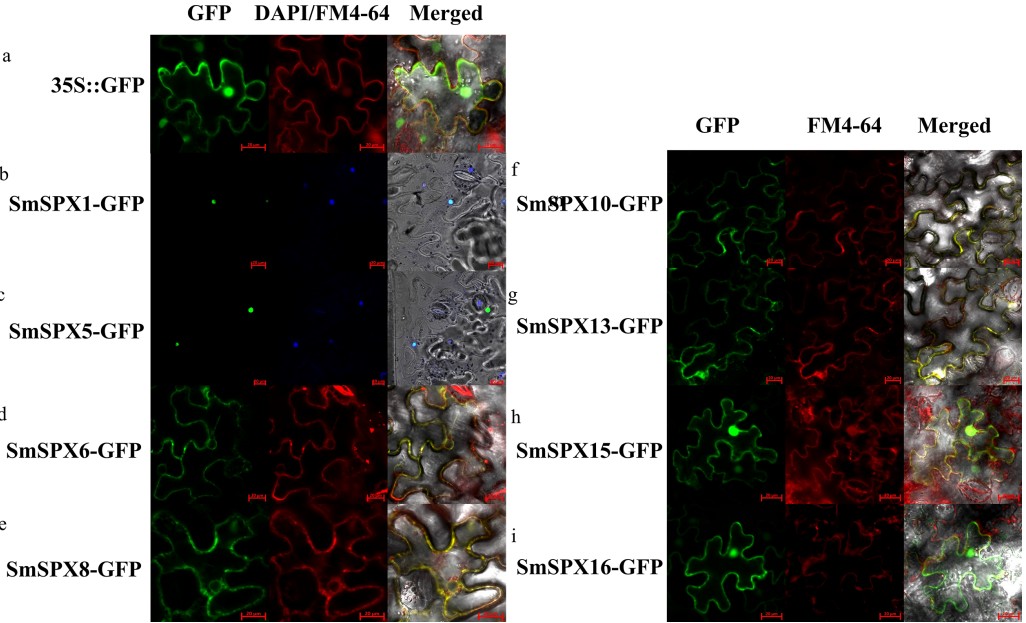

**Figure 3** Subcellular localization of *SPX domain-containing* proteins (bars = 20 μm). SmSPX-GFP fusion proteins were transiently expressed in tobacco leaves, and their localization was determined using confocal microscopy. (A) *35S::* GFP is located in the nucleus, cytoplasm and plasma membrane; (B–C) SmSPX1 and SmSPX5 are located in the nucleus; (D–E) SmSPX6 and SmSPX8 are located on the membrane system; (F–G) SmSPX10 and SmSPX13 are located on the cell membrane; (H–I) SmSPX15 and SmSPX16 are expressed in both the nucleus and the cell membrane. Red indicates the location of plasma membrane; green indicates the position of SmSPX proteins; dark blue represents the location of the nucleus; yellow merged represents the co-localization of plasma membrane and SmSPX proteins; Light blue represents the nucleus co-localized with SmSPX proteins in the nucleus.

genes expression levels were surveyed in the transcriptome data. According to RNA-seq data, the expression level of most *SPX domain-containing* genes increased at 1 day and changed significantly at 1~5 days by low phosphate, and differed at 5~10 days in the two varieties. The expression levels of *SmSPX4*, *SmSPX5*, *SmSPX1*, *SmSPX3*, and *SmSPX14* between the two varieties were differences (Fig. 5). *SmSPX4*, *SmSPX5*, *SmSPX1* and *SmSPX14* expression level in V8 variety was higher than 41 variety at 1 day and 5 days, and there was no significant difference in expression level at 10 days, this difference may be key genes in response to phosphate deficiency stress.

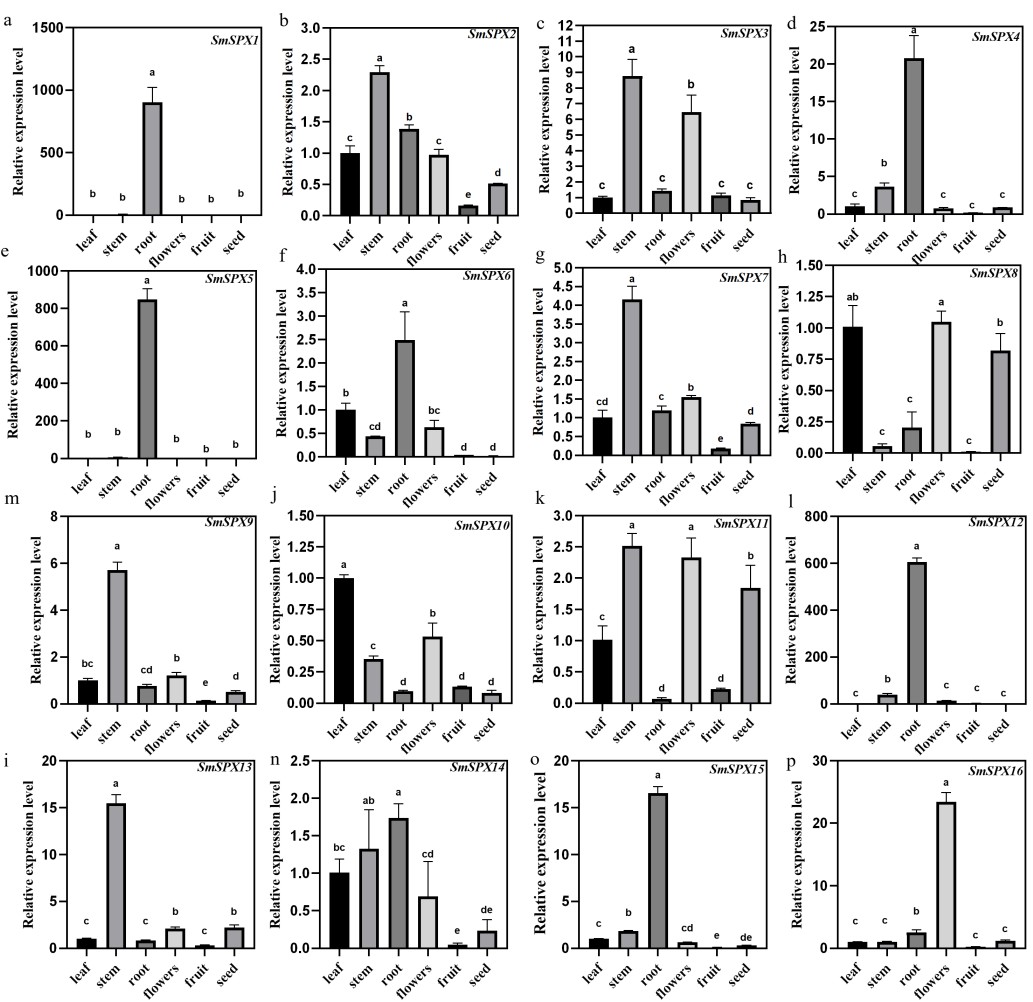

**Figure 4** **The relative expression level of 16 *SPX domain-containing* genes in the leaves, stems, roots, flowers, fruits and seeds.** (A–P) The relative expression of *SmSPX1-SmSPX16* in different plant tissues. The relative expression level was calculated using the method of $2^{-\Delta\Delta}$Ct. The relative mRNA levels of the leaves were used for the reference. The values are means ± SD ($n = 3$). An asterisk (*) represents significance at $p < 0.05$ comparing with reference.

## Expression analysis of *SPX domain-containing* genes in response to IAA

To further identify the biological reaction pathways in which *SPX domain-containing* genes involved, cis-elements in the promoters were analyzed (Fig. S4). Six main classes of cis-acting elements were found: including photo-responsive elements, coercion of the corresponding components, hormone-responsive elements, circadian response elements, seed developmental response elements, and *MYB* binding sites, *etc*. It is worth noting that *SPX domain-containing* genes have many hormone-responsive elements and photo-responsive elements, which may be a key way for its response to stress to regulate plant development. Considering *SPX domain-containing* genes and auxin are both members that play a main role under phosphate deficiency stress, we analyzed the relationship

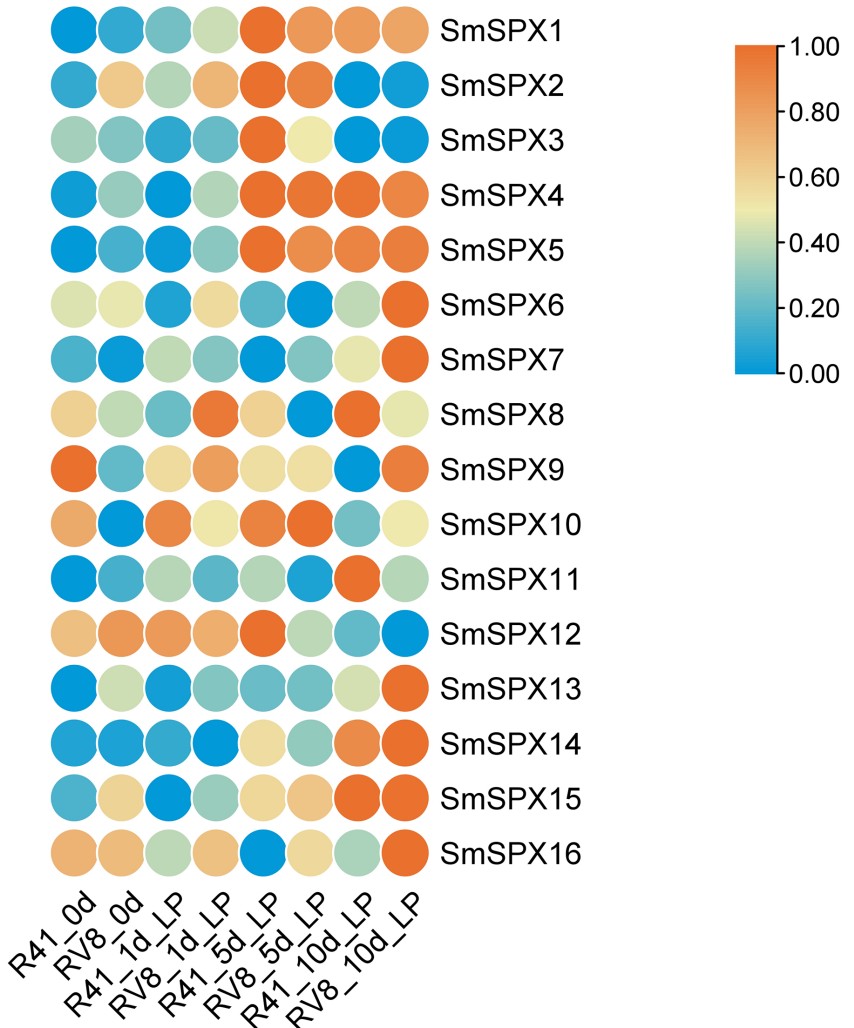

**Figure 5** **Expression levels of 16 *SPX domain-containing* genes in two multi-generation self-bred varieties eggplant seedlings under low phosphorus treatment on different days.** Different colors indicate different expression level, the highest in red and the lowest in blue.

between *SPX domain-containing* genes and auxin under phosphate deficiency stress. We added 0.2 mg/L IAA to phosphate starvation treatment on 4-week-old eggplant seedlings in hydroponic culture, and detected the expression patterns of 16 *SPX domain-containing* genes at 1 day, 5 days, and 10 days of phosphate starvation, respectively (Fig. 6). The results showed that the expression of *SPX domain-containing* genes under phosphate starvation treatment combination with IAA was significantly different from that of phosphate starvation treatment alone. *SmSPX1*, *SmSPX4*, *SmSPX5*, and *SmSPX15* were significantly induced by IAA under phosphate deficiency stress, especially at 10 days.

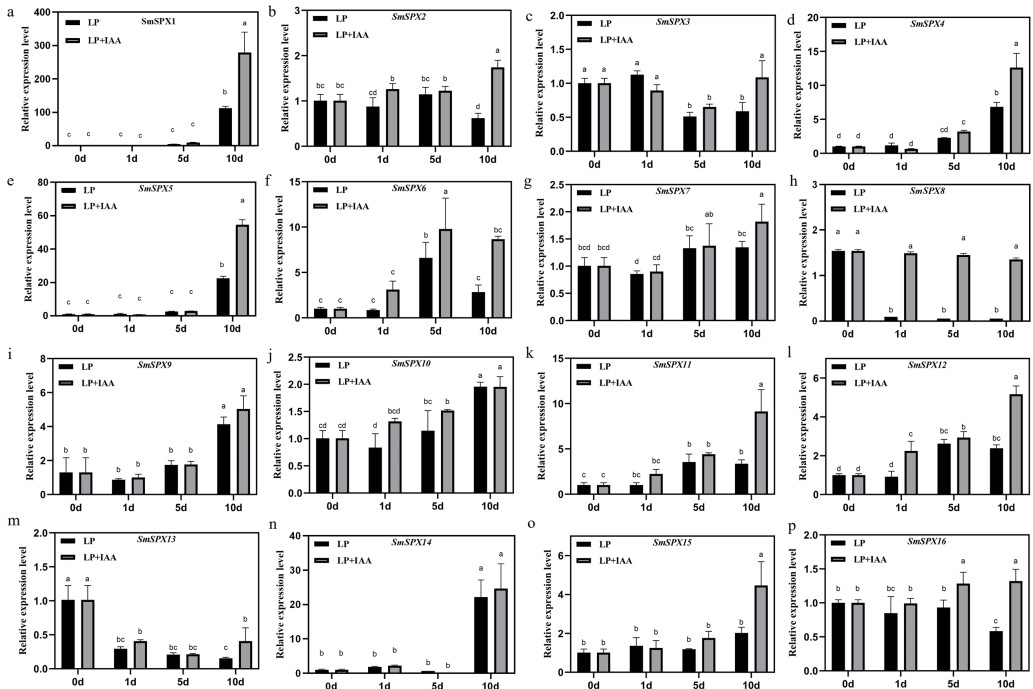

**Figure 6** Gene expression levels of 16 kinds of *SPX domain-containing* genes in roots at 1 day, 5 days and 10 days after low phosphorus and low phosphorus treatment with IAA. (A–P) The relative expression of *SmSPX1-SmSPX16* in different treatment, black indicates phosphorus deficiency treatment, gray indicates phosphorus deficiency when adding IAA treatment. The relative expression level was calculated using the method of $2^{-\triangle\triangle Ct}$. The relative mRNA levels of the leaves were used for the reference. The values are means ± SD ($n = 3$). An asterisk (*) represents significance at $p < 0.05$ comparing with reference.

## DISCUSSION AND CONCLUSIONS

Phosphate deficiency has always been an important factor affecting plant development, and plants produce corresponding physiological and biochemical reactions to increase the way to obtain P from the soil or reduce their unnecessary nutrient consumption. Under phosphate deficiency condition, the stress response of plants is mainly regulated by the P signaling pathway, and the *SPX domain-containing* genes play an irreplaceable role as an important component of the P signaling response. Therefore, evaluating the biological information of SPX protein can better understand the signaling mechanism of eggplant under phosphate deficiency and contribute to future genetic breeding.

In our study, we identified 16 *SPX domain-containing* genes in eggplant, which are very similar to the SPX family members (19) in tomato (*Li, You & Zhao, 2021*). Three segments of repetitive gene pairs were found on four chromosomes of eggplant, indicating that these genes underwent strong selection during the evolution of eggplant (Fig. 1B). The function of SPX domain proteins could be determined by the evolutionary relationships among different species (Fig. 1C). The isograms of eggplant with *Arabidopsis* and tomato showed that most of these genes are conserved during evolution, similar to those reported in *Arabidopsis* (*Duan et al., 2008*). The majority of homologous genes between *Arabidopsis*
and eggplant are found to be located on chromosomes 1 and 4 of *Arabidopsis*, as well as chromosomes 2 and 8 of eggplant. This suggests that *SPX* s changed throughout evolution; however, some *SPX* genes have remained relatively conserved. The induced expression of *SmSPX* genes under low phosphorus conditions also provides evidence for its role in resisting phosphorus deficiency in eggplant. We identified the subcellular localization of *SmSPX1* and *SmSPX5* and found that they are both located in the nucleus. *SmSPX6* and *SmSPX8* are located to the membrane system. *SmSPX10* and *SmSPX13* localized to the plasma membrane. *SmSPX15* and *SmSPX16* expressed both in nuclei and cell membranes (Fig. 3). *SmSPX1*, *SmSPX4*, and *SmSPX5* belong to the same subfamily, and specifically expressed in roots. The expression trend of *SmSPX1*, *SmSPX4*, and *SmSPX5* in different eggplant varieties is also consistent under phosphate deficiency situation, indicating that this subfamily may play an important role in eggplant phosphate deficiency stress (Figs. 4 and 6, Fig. S3). However, the expression level of *SmSPX1* in anti-phosphate deficiency eggplants variety (V8) is higher than that in phosphate deficiency eggplants variety (41) at the first and fifth day under phosphorus deficiency. From an evolutionary perspective, *SmSPX1*, *SmSPX4*, and *SmSPX5* are similar to *AtSPX1* and *AtSPX3*. Previous study has reported that overexpression of *AtSPX1* increases the transcriptional levels of *ACP5*, *RNS1*, and *PAP2* under phosphorus sufficient and phosphorus deficient conditions in *Arabidopsis*, indicating that *AtSPX1* has a potential transcriptional regulatory role in response to phosphorus starvation (*Duan et al., 2008*). *OsSPX1* inhibits the formation of *OsPHR2* dimer in rice, and its overexpression suppresses the expression of phosphate starvation-induced gene and disrupts its function in the phosphorus deficiency pathway (*Liu et al., 2010*; *Wang et al., 2009*). Nine *GmSPX* members have been identified in soybeans. Overexpression of *GmSPX1* reduces the total P concentration in plants, alters root hair morphology, and inhibits both root hair elongation and quantity elongation (*Zhang et al., 2016*). Overexpression of *GmSPX3* gene enhances P content in surface soil and root hairs, as well as upregulates transcription of seven genes involved in phosphorus hunger response in soybean root hairs (*Yao, Tian & Liao, 2014*). The expression levels of all the SPX subfamily identified in maize are increased under low phosphorus deficiency except *ZmSPX3*, among which *ZmSPX4.1* and *ZmSPX4.2* were the most obvious. Moreover, *ZmSPX3* and *ZmSPX4.2* can regulate phosphorus deficiency in a *ZmPHR1*-mediated manner (*Xiao et al., 2021*). Taken together the *SPX* subfamily members from different plant species seem to have a unified function. As shown in (Fig. 6 and Fig. S3), *SmSPX1* was most significantly up-regulated after low phosphorus deficiency, we speculate that this expression difference of *SmSPX1* between V8 and 41 variety is one of the reasons for the higher anti-phosphate deficiency stress of V8, *SmSPX1* may play a positive regulatory role in eggplant adaptation to phosphorus starvation. *SmSPX6* and *SmSPX8* are members of *SmSPX-MFS* subfamily, mainly expressed in the membrane system, which were similar to *AtPHT5;1* and *AtPHT5;3* (Fig. 3). Under phosphate deficiency stress, the expression level of *SmSPX6* was increased, while *SmSPX8* was significantly decreased. A similar situation also exists in the root system of *Arabidopsis*, *AtPHT5;1* was up-regulated in response to P deficiency, but *AtPHT5;3* was down-regulated (*Liu et al., 2016b*). Overexpression of *AtPHT5;3* results in mis-regulation and growth retardation of the *PSR* gene as a result of large amounts of P sequestered to the vacuole (*Liu*
*et al., 2016b*). There are three vacuolar phosphorus transporters (*OsSPX-MFS1-3*) in rice, and they were mainly expressed in shoots. *OsSPX-MFS1* and *OsSPX-MFS3* were inhibited by P deficiency, while *OsSPX-MFS2* was induced (*Guo et al., 2023*; *Wang et al., 2012*). The mutants of *osspx-mfs1* and *3* showed lower vacuolar P concentration, while, *OsSPX-MFSs* overexpressed plants showed higher vacuolar P accumulation. From the evolutionary point, the function of *SmSPX6* and *SmSPX8* may be conserved with the homologous genes in *Arabidopsis*. *SmSPX8* may be the most important vacuolar phosphorus transfer protein, because its expression is severely reduced in phosphorus deficiency. *SmSPX10-14* and *AtPHO1* belong to the SPX-EXS subfamily genes. The function of *AtPHO1* is mainly phosphate transport in cells, and phosphoric acid deficiency induces *AtPHO1* expression (*Wang et al., 2004*). *AtPHO1* homologous gene *OsPHO1;2* in rice plays a key role in the transfer of P from roots to shoots (*Secco, Baumann & Poirier, 2010*). In eggplant, *SmSPX10* is abundant in leaves, *SmSPX12* is abundant in roots, and *SmSPX13* is abundant in stems (Fig. 3). The expression of these genes in specific locations may ensure the unhindered transport of phosphorus among organs. *SmSPX11* is abundant in reproductive organs, which may be an important factor in determining the flowering and fruit setting of eggplant. However, *SmSPX14* seems to be expressed in all sites except fruits, which may indicate that it is essential for its P transport function in plants. Although the expression pattern of *SmSPX14* in two varieties of eggplant seedlings is resemblance, the expression level of *SmSPX14* in V8 variety is twice than that of 41 under phosphate deficiency at one day (Fig. S3). We speculate that *SmSPX14* might pay a role on the higher anti-phosphate deficiency stress of V8.

*SmSPX15* and *SmSPX16* have a RING domain, which is similar in structure to *AtSYG1* reported in *Arabidopsis*. Previous studies have reported that AtSYG1 could polyubiquitination of PHR1 *in vitro*, resulting in SPX-PHR1 complex untangling and activation of downstream PSR gene expression (*Park et al., 2023*). Here, *SmSPX15* was induced under phosphate deficiency stress, nevertheless, the expression level of *SmSPX16* remained almost unchanged. This may be related to the expression specificity, of *SmSPX16*, which may play other functions in flowers organs (Fig. 3).

Many hormone-responsive elements were found in the promoters of *SPX domain-containing* genes, indicating that plant hormones may play an important role in regulating the expression of *SPX domain-containing* genes. Moreover, *AtPHO1;H1* and *AtPHO1;H10* have been found to be up-regulated by auxin under phosphate deficiency in *Arabidopsis* (*Ribot, Wang & Poirier, 2008*). Here, we found that the expression of *SmSPX1*, *SmSPX4*, *SmSPX5* and *SmSPX15* were also induced by auxin under phosphate deficiency (Fig. 6). These genes cover all subfamilies, fully demonstrating the close relationship between auxin and SPX *domain-containing* in eggplant responding to phosphate deficiency. These findings are worthy of further study and provide theoretical guidance for broadening the molecular mechanism of plant response to phosphate deficiency stress in the future.

### Abbreviations

| | |
|---|---|
| **SPX** | SYG1, PHO81, and XPR1 |
| **SPX-EXS** | SPX-ERD1/XPR1/SYG1 |

| **SPX-MFS** | SPX-Major Facilitator Superfamily |
| **SPX-RING** | SPX-Really Interesting New Gene |
| **P** | Phosphorus |
| **PSI** | Phosphate starvation induced |
| **aa** | Amino acids |
| **MWs** | Molecular weights |
| **pI** | Isoelectric points |
| **CDS** | Coding sequence |
| **DAPI** | $4', 6$-diamidino-2-phenylindole |
| **FM4-64** | N-(3-triethylammom iumpropyl)-4-(p-diethylam inophenylhexatrienyl). |

### Funding

This work was supported by the National Natural Sciences Foundations of China (No. 32172556), the Key Project of Shandong Provincial Natural Science Foundation (ZR2020KC039), and the Shandong Provincial Key Research and Development Program (2022LZGC009). The funders had no role in study design, data collection and analysis, decision to publish, or preparation of the manuscript.

### Grant Disclosures

The following grant information was disclosed by the authors:
National Natural Sciences Foundations of China: No. 32172556.
Key Project of Shandong Provincial Natural Science Foundation: ZR2020KC039.
Shandong Provincial Key Research and Development Program: 2022LZGC009.

### Competing Interests

The authors declare there are no competing interests.

### Author Contributions

- Li Zhuomeng conceived and designed the experiments, performed the experiments, analyzed the data, prepared figures and/or tables, and approved the final draft.
- Tuo Ji conceived and designed the experiments, prepared figures and/or tables, and approved the final draft.
- Qi Chen analyzed the data, prepared figures and/or tables, and approved the final draft.
- Chenxiao Xu performed the experiments, prepared figures and/or tables, and approved the final draft.
- Yuqing Liu performed the experiments, prepared figures and/or tables, and approved the final draft.
- Xiaodong Yang conceived and designed the experiments, authored or reviewed drafts of the article, and approved the final draft.
- Jing Li conceived and designed the experiments, authored or reviewed drafts of the article, and approved the final draft.
- Fengjuan Yang conceived and designed the experiments, authored or reviewed drafts of the article, and approved the final draft.

## DNA Deposition

The following information was supplied regarding the deposition of DNA sequences:

The sequences are available at GenBank: PRJNA1030332, SAMN39278703–SAMN39278741.

## Data Availability

The bar chart statistics are available in the Supplemental Files.

The transcriptome raw data is available at figshare: Li, Zhuo Meng (2024). Eggplant transcriptome raw data. figshare. Dataset. https://doi.org/10.6084/m9.figshare.25388260.v10.

## Supplemental Information

Supplemental information for this article can be found online at http://dx.doi.org/10.7717/peerj.17341#supplemental-information.

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
