# Peer review of "Genome-wide identification and characterization of SPXdomain-containing genes family in eggplant"

_PeerJ, doi:10.7717/peerj.17341_

## Round 0.1 · original submission · Major Revisions

Dear Authors
The manuscript cannot be accepted for publication in its current form. The manuscript needs substantial revision to meet the standards of the journal. The authors are invited to revise the paper considering all the suggestions made by the reviewers, including one who recommended rejecting the manuscript. Please note that requested changes are required for publication.

In addition, there are significant concerns about the manuscript's grammar, usage, and overall readability. We, therefore, request that you revise the text to fix the grammatical errors and improve the overall readability of the text.

With Thanks

**Language Note:** The Academic Editor has identified that the English language must be improved. PeerJ can provide language editing services - please contact us at [email protected] for pricing (be sure to provide your manuscript number and title). Alternatively, you should make your own arrangements to improve the language quality and provide details in your response letter. – PeerJ Staff

Reviewer 1 ·

Basic reporting

The writing part doesn't conform to professional standards of expression.
1. Line 13:“Zhuomeng Li1 , ” and “Xiaodong Yang4, Jing ” Space should be removed.
2. Line 43, 150, 151, 152, 161, 163, 172, 174, 188, 189, 199, 200, 216, 217, 231, 236, 252, 271, 288, 309, etc: The beginning does not need to be indented.
3. Line 248: "The predicted molecular weight ranged from 29.28 to 183.38, and the isoelectric point ranged from 4.69 to 9.41." Replace "29.28 to 183.38" by "29.28KDa to 183.38KDa".
4. Lines 276-279: "The second subfamily contains SPX and MFS domains, including 3 Arabidopsis proteins, 4 SlSPX-MFS and 4 SmSPX-MFS. The third subfamily of SPX and EXS domains consist of 11 Arabidopsis genes, 8 SlSPX-EXS and 5 SmSPX-EXS. " The expression of "genes" and "proteins" is confusing. You need to replace "4 SlSPX-MFS and 4 SmSPX-MFS" by "4 SlSPX-MFSs and 4 SmSPX-MFSs" and "8 SlSPX-EXS and 5 SmSPX-EXS" by "8 SlSPX-EXSs and 5 SmSPX-EXSs".
5.Lines 447-448: Figure 4. "The expression profiles of 16 SPX domain-containing genes in the leaves, stems, roots, flowers, fruits and seeds of 4–week–old eggplants." "4–week–old eggplants" ???

Experimental design

Experimental designs are not clear.
1. Figure 4. "The expression profiles of 16 SPX domain-containing genes in the leaves, stems, roots, flowers, fruits and seeds of 4–week–old eggplants." "4–week–old eggplants" ???
2. Line 321: "To identify the key genes under phosphate deficiency stress, we made root transcriptomes of two varieties with different phosphate deficiency tolerance treating with low phosphate for 1 day, 5 days, and 10 days." RNA-seq raw data should be submitted to the database and obtain an access number.
3. why are you selected 8 (SmSPX1, SmSPX5, SmSPX6, SmSPX8, SmSPX10, SmSPX13, SmSPX15 and SmSPX16) members for subcellular localization? why not analyze all?
4."flowers" or "flower buds"?

Validity of the findings

no comment

Additional comments

Na

Reviewer 2 ·

Basic reporting

About the manuscript " Identification of SPX domain-containing genes family in the eggplant genome and expression under phosphate deficiency with indoleacetic acid ", the topic is clear. And the structure, figures, tables are acceptable.

Experimental design

no comment

Validity of the findings

no comment

Additional comments

There are some essential problems should be addressed by authors, which are listed below.

1.Why did the author choose Arabidopsis, Solanum lycopersicum and Solanum melongena to construct the evolutionary tree? What are the evolutionary relationships among other classical species? For example, rice, etc. Did the author consider the differences and evolutionary relationships between dicotyledonous and monocotyledonous plants?
2.Could you elaborate further on the evolutionary history in relation to their proposed functions?
3.What influenced the authors' choices for V8 and NO.41 Multigenerational inbred lines?
4.For Subcellular localization, are the SPX gene and software prediction results consistent? It has been reported that some SPX proteins are found in the cytoplasm and can enter the nucleus to interact with other proteins. In this study, the author investigated the subcellular localization of SPX family proteins using the tobacco system. Can the author's diagram accurately depict the protein localization? Should organelle-tagged tobacco systems or protoplast systems be considered for relocalization of proteins with undefined localization?
5.SPXs may be affected by many hormones or other factors. Why did the author choose IAA? Is there any comparison with others, and after comparison, it is found that SPX is more affected by IAA?
6.The discussion is not too powerful. I suggest authors restructure the discussion section comprehensively by making constructive comparisons with previous findings and highlighting how different or comparable their findings are with published findings.
7.The abbreviation of phosphorus should be unified, Pi or P.

·

Basic reporting

Comments

Abstract
Not to start a sentence with number ie 16 please change to Sixteen

Discussion and conclusion
The term outside world in first paragraph is not scientific, please use a more suitable word term.
Even though the study and findings are comprehensive, the discussion is not sufficient. First, there should be comparison with findings from other plants beside Arabidopsis. In some places, only the results were stated without a proper discussion of the significance and not compared with more recent findings from Arabidopsis and other species as SPX domain containing genes/ proteins have been studied in several species. The Introduction is too extensive, possible some relevant information can be used as supporting information in the Discussion part.

General
Throughout the text, please give space before bracket for putting citations. It is noted the numbering of the genes and the presentation of results had been arranged based on the subfamilies ie grouping based on subfamilies for better organisation and clarity

Experimental design

Good for studying SPX domain containing gene families in egg plant based on gene structure, expression profile and response to phosphate deprivation and auxin treatment

Validity of the findings

Yes valid based on the experimental design, the different methods used and the statistical analysis

Reviewer 4 ·

Basic reporting

The English of the manuscript should be improved globally.

Experimental design

no comment

Validity of the findings

no comment

Additional comments

In this study, authors systematically conducted the identification and expression analysis of SPX domain-containing genes family in eggplant. The results obtained were valuable for better understanding the roles of SPX domain-containing genes family in the response to low phosphate stress in eggplant. However, revisions may need before this manuscript could be considered for publication:
1. The manuscript writing quality is worse and it should be revised comprehensively for grammar, spelling, and format mistakes (also need to check the spelling of author names).
2. The title should be revised, which is not concise and precise.
3. In introduction, the references related to auxin signaling under Pi deficiency are not sufficient. And the previous work had reported that PHR1 is transcriptionally regulated by auxin (Huang et al., 2018 Plant Physiol).
4. The order of 0, 1, 5,10 D LP is confused in Figure 5, which should be re-edited.

---

## Round 0.2 · Minor Revisions

Dear Authors

The manuscript still needs a minor revision before it can be reconsidered for publication. The authors are invited to revise the paper, considering all the reviewers' suggestions.

With Thanks

Reviewer 2 ·

Basic reporting

The manuscript is improved much. My major concerns are partially addressed.

Experimental design

no comment

Validity of the findings

no comment

Additional comments

no comment

Reviewer 4 ·

Basic reporting

no comment

Experimental design

no comment

Validity of the findings

no comment

Additional comments

Although the quality of the new version is improved compared to original manuscript, multiple mistakes also could be found throughout the manuscript. The manuscript still should be revised globally and carefully.
Some points:
1. The names of the authors should be located before the authors’ affinities.
2. Except first Solanum melongena, the other should be written as “S. melongena”.
3. Line 21: “4” should be revised as “four”.
4. Line 116-118: ARF7 and ARF19 regulate PHR1 gene not PHR1 protein. The reference (Huang et al. 2018) is cited at a wrong position.
5. Line 263: the subtitle “Subcellular localization of SPX domain-containing genes” should be revised as “Subcellular localization of SPX domain-containing proteins”.
6. The list of the Reference should be checked one by one and there are too many mistakes, including the spelling of authors and journals etc.
7. The Figure legends also need to revise. Each Figure should have a title.
8. Although Figure 5 was re-edited, the data are not consistent with the original manuscript. For example, the expression pattern of smSPX7, smSPX8 was obviously different from the data in the former version of Figure 5.

---

## Round 0.3 · accepted · Accept

Dear Authors,

I am pleased to inform you that the manuscript has improved after the last revision round and can be accepted for publication.

Congratulations on accepting your manuscript, and thank you for your interest in submitting your work to PeerJ.

With Thanks